# Efficacy of Prolonged-Release Melatonin 2 mg (PRM 2 mg) Prescribed for Insomnia in Hospitalized Patients for COVID-19: A Retrospective Observational Study

**DOI:** 10.3390/jcm10245857

**Published:** 2021-12-14

**Authors:** Carolina Bologna, Pasquale Madonna, Eduardo Pone

**Affiliations:** UO Pneumologia Subintensiva COVID Ospedale del Mare ASL Na1, 80147 Naples, Italy; linomadonna@libero.it (P.M.); eddi22@alice.it (E.P.)

**Keywords:** COVID, SARS-CoV, prolonged-release melatonin 2 mg, delirium, insomnia

## Abstract

Background: we have observed the effect of insomnia treatment in clinical and prognostic differences of patients admitted for COVID-19 pneumonia in respiratory sub-intensive units that were administered a prolonged-release melatonin 2 mg (PRM 2 mg) therapy versus a group of patients out of therapy. Materials and Methods: We evaluated 40 patients on prolonged-release melatonin 2 mg (PRM 2 mg) therapy versus a control group of 40 patients out of therapy. Results: patients in the PRM 2 mg group had a shorter duration of therapy with non-invasive ventilation (5.2 ± 3.0 vs. 12.5 ± 4.2; *p* < 0.001), with a shorter stay in sub-intensive care (12.3 ± 3.2 vs. 20.1 ± 6.1; *p* < 0.001), and, therefore, a shorter overall duration of hospitalization (31.3 ± 6.8 vs. 34.3 ± 6.9 *p* = 0.03). In addition, a lower incidence of delirium was found (2.2 ± 1.1 vs. 3.3 ± 1.3; *p* < 0.001). Conclusions: A significant increase in sleep hours and a reduction in delirium episodes occurs in hospitalized insomniac patients treated with PRM 2 mg, compared to untreated patients. Based on these preliminary results, we can assume that there are benefits of prolonged-release melatonin 2 mg in COVID-19 therapy.

## 1. Introduction

Numerous scientific studies published in the last year (MEDLINE and EMBASE) highlight the likely benefits of melatonin in the treatment of COVID-19 [1,2,3]. An excessive inflammatory response, cytokine storms, and the subsequent progression to acute lung injury (ALI)/acute respiratory distress syndrome (ARDS) constitute the pathology of COVID-19. Melatonin, a known anti-inflammatory and antioxidant molecule, is protective against ALI/ARDS caused by viral and other pathogens. Melatonin is effective in sub-intensive care patients by reducing vessel permeability, anxiety, for hypnotic use, and improving sleep quality, which could also be beneficial for better clinical outcomes in COVID-19 patients. Furthermore, melatonin has a high safety profile [1]. There is significant data showing that melatonin limits virus-related diseases, such as diabetes and vascular and metabolic complications. It reduces neurological sequelae, acute and chronic, and can enhance the protection of vaccination [4]. Cardinali [4], in his review published in “Disease”, defines melatonin as the “silver bullet” of the therapy for COVID-19COVID-19. More experiments and clinical studies are needed to confirm this hypothesis [4].

Insomnia is the most common sleep disorder encountered in the geriatric clinic population, frequently characterized by the subjective complaint of a difficulty to fall or maintain sleep, or nonrestorative sleep, producing significant daytime symptoms, including the difficulty to concentrate and mood disturbances [5]. In Italy, since 2013, there is only one product registered as a prolonged-release melatonin drug at a dose of 2 mg, approved for the insomnia treatment characterized by a poor quality of sleep in people aged ≥ 55. The maximum concentration allowed for melatonin-based food supplements is 1 mg and there is a lack of data that proves the efficacy and safety of the different commercial products [6,7]. This is of particular importance, given the high variability in content compared to the level of melatonin concentrations and the presence of the contaminants [8]. PRM 2 mg mimicked the physiological release of melatonin by releasing melatonin gradually and acting on melatonin receptors. PRM 2 mg has been shown to be effective in improving sleep latency (SOL) and total sleep time (TST) without altering the physiological sleep structure [9,10,11,12,13]. PRM 2 mg has been shown to be well tolerated and does not present side-effects, such as a hangover; nocturnal confusion and falls; negative effects on next-day cognitive performance; rebound insomnia; tolerance; and dependency [6,14,15]. It should also be remembered that both European guidelines and an Italian consensus indicate PRM 2 mg as the first therapeutic choice in insomniac subjects over the age of 55 years, for up to 13 weeks [8,16,17]. The aim of the study was to investigate the effect of insomnia treatment with prolonged-release melatonin 2 mg in clinical and prognostic differences of patients admitted for COVID-19 pneumonia in respiratory sub-intensive units versus a group of patients that were not administered that therapy: the goal is to demonstrate that improved sleep management reduces episodes of delirium, improves patient compliance to ventilation, and generally improves prognosis.

## 2. Materials and Methods

This retrospective observational study aims to evaluate the benefits and efficacy of PRM 2 mg for the treatment of insomnia in COVID-19 patients compared to a group without therapy. This study focused on patients with COVID-19-related pneumonia, hospitalized in a sub-intensive unit, through the following serial measurements:Total number of hours of sleep ensured for the patient;Duration of non-invasive ventilation;Duration of the sub-intensive stay;Overall hospitalization duration;Incidence of delirium, an acute disturbance of consciousness with inattention accompanied by a change in cognition, or perceptual disturbance that fluctuates over time [9].

### Authorization of the Ethics Committee Opinion Expressed in the Session of 28 April 2021 Prot. 275\CE22-2021 ASL Na1

This study included patients hospitalized in the COVID Sub-intensive Care Unit (Ospedale del Mare, ASL NA1 Centro, Naples, Italy), with a clinical diagnosis of SARS-CoV-2 virus infection confirmed by PCR or other approved diagnostic methodology; for COVID-19-induced pneumonia documented by chest X-ray or CT with evidence of pulmonary infiltrates; in autonomous breathing or requiring oxygen supplementation or non-invasive mechanical ventilation upon admission; and with a preserved swallowing function.

This study excluded patients that needed invasive mechanical ventilation upon admission; presented hypersensitivity to the active ingredients and/or to any excipient present; severe hepatic and renal insufficiency; severe heart disease; severe dementing syndrome; or those who were in a comatose state, presenting conditions of advanced terminality.

In this retrospective observational study, 80 patients were included: 40 patients in the melatonin PRM 2 mg group therapy versus 40 in the control group. The study population included male and female subjects, diversified by age groups and hospitalized for COVID-19-related pneumonia.

The patients’ characteristics were reported as percentages or summarized and tabulated by means and standard deviation (SD). The endpoints of interest were described by calculating their median and interquartile (IRQ) range, or mean and standard deviation (SD), when found to have a non-normal or normal distribution, respectively, by applying the Shapiro–Wilk test to assess if the melatonin administration was effective in reducing the duration and the severity of the pathology. The endpoints of interest collected from the melatonin-treated group were compared to those obtained from the control group by means of parametric and non-parametric tests for normal and non-normal variables, respectively, and namely the two-sample t-test for unpaired data or the Mann–Whitney test. For the sake of completeness, primary endpoints (number of sleep hours and number of episodes of delirium) were also analyzed using the chi-square test. The test results were regarded as significant if *p* < 0.05. All statistical analyses were performed using OriginPro (OriginLab, Northampton, MA, USA).

## 3. Results

In this study, 80 patients were observed and analyzed in 2 different groups: 40 patients (23 males and 17 females, with a mean age of 71.6 years) in the group treated with PRM 2 mg and 40 patients in the control group (we considered 40 patients, 23 males and 17 females, with a mean age of 71.8 years) (Figure 1).

Clinical parameters, such as the body mass index and blood chemistry for kidney and liver function, the inflammation indices, and the blood count, were homogeneous in both groups, as shown in Table 1.

The group of patients treated with PRM 2 mg had a median of 6 h of sleep, compared with a median of 5 h in the untreated group. The distribution of the number of hours of sleep was not normal; therefore, the Mann–Whitney test was used to determine the statistical significance (average total sleep time: 5.5 ± 0.8 vs. 4.5 ± 1.2; *p* < 0.001) (Figure 2). There was also a statistically significant reduction of episodes of delirium in the group treated with melatonin PRM 2 mg: a median of 2.2 episodes in the median versus 3.3 episodes in the untreated group with a statistical significance (*p* < 0.001) (Figure 3).

Table 2 shows the comparison data between the two groups for the total duration of hospitalization and days of non-invasive ventilation. The distribution of data for the days of hospitalization and for the days spent in sub-intensive therapy was normal; therefore, to determine the statistical significance, the two-sample t-test was used. The mean and the respective standard deviation (Std. Dev.) provided valid qualitative data to describe the distribution of the population. In the PMR 2 mg group, the mean duration of sub-intensive hospitalization was 12.3 ± 3 days versus 20.1 ± 6.1 days in the untreated group (*p* < 0.001).

The distribution of data for the days spent with non-invasive ventilation (NIV) and high-flow ventilation was not normal; therefore, to determine the statistical significance, the Mann–Whitney test was used. In addition to the mean (and the respective standard deviation), the median and interquartile range (IQR) values were entered as they were a more informative way to describe the population distribution. The duration of non-invasive ventilation was also significantly shorter in the PRM 2 mg group compared to the untreated group, with a mean of 5.2 ± 3.0 days versus 12.5 ± 4.2 days (*p* < 0.001). On the other hand, there was no significant difference in the duration of ventilation with high flows. 

Finally, we compared the number of patients transferred from sub-intensive to intensive care due to the clinical worsening. Out of the patients who, unfortunately, had an unfavorable outcome in the group treated with PRM 2 mg, only 7.5% versus 15% of the untreated group were transferred to intensive care due to clinical worsening, and 15% of the untreated group died versus 7.5% of the treated group (Figure 4). 

No adverse events related to the drug or the need to discontinue therapy were observed, except in four patients who had an unfavorable evolution of pneumonia for which they underwent orotracheal intubation (IOT) and were transferred to the sub-intensive unit.

## 4. Discussion

Cognitive impairment delirium [9] can be the first presenting symptom of patients with the SARS-CoVinfection, complicating the management of these patients in sub-intensive units, making non-invasive ventilation difficult and worsening the overall prognosis of patients, while prolonging the duration of hospitalization and often causing the failure of ventilatory management by rendering it necessary to intubate and transfer the patient to the ICU [10,11]. Melatonin received considerable scholarly attention during the pandemic. The first article, published by Baller [18], was followed by a series of studies on the anti-inflammatory role of the hormone [19,20,21,22,23,24]. The artificial light of sub-intensive therapy makes sleep fragmented and unstructured, and melatonin levels are lower in the elderly [25]. Many hospitalized patients were already receiving melatonin therapy. Given its anti-inflammatory action, we decided not to suspend melatonin, but, rather, to compare a group of treated patients with an untreated group. We enrolled patients with similar clinical characteristics in the two groups with homogeneous samples. It is clear that patients who showed a greater number of hours of sleep were calmer, more cooperative, and with fewer cases of delirium. This contributed, in our opinion, to the successful ventilation and rapid weaning to high flow therapy which improved the prognosis of these patients. Furthermore, only 7.5% of patients treated with prolonged-release melatonin 2 mg were transferred to the ICU and died during the hospital stay, while, in the untreated group, the percentage of patients who were transferred to the ICU with fatal outcomes doubled, rising to 15%. No adverse events related to the drug or the need to discontinue therapy were observed, except in four patients who had an unfavorable evolution of pneumonia, for which they underwent Oorotracheal intubation (IOT) and were transferred to the intensive care unit.

## 5. Conclusions

Based on these preliminary results, we can assume that there are benefits to administering PRM melatonin 2 mg in COVID-19 therapy, such as an improved sleep quality, reduced delirium risk, as well as improved disease prognosis, probably related to the melatonin’s potential anti-inflammatory and antioxidant effects. However, more clinical studies are needed to confirm this hypothesis.

## Figures and Tables

**Figure 1 jcm-10-05857-f001:**
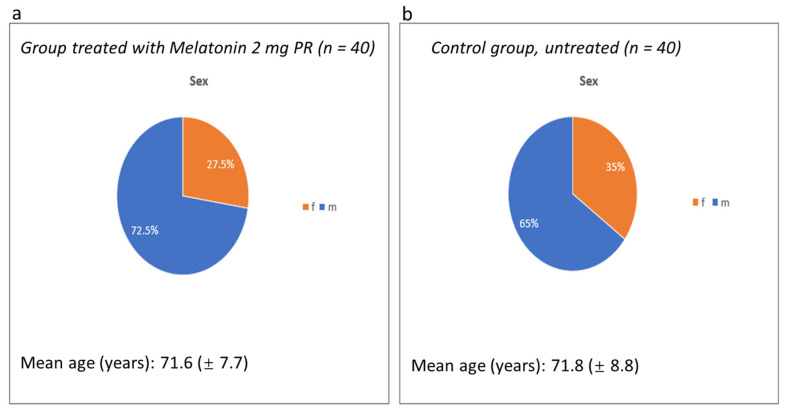
(**a**): 40 patients (23 males and 17 females, with a mean age of 71.6 years) in the group treated with PRM 2 mg. (**b**): 40 patients in the control group (we considered 40 patients, 23 males and 17 females, with a mean age of 71.8 years).

**Figure 2 jcm-10-05857-f002:**
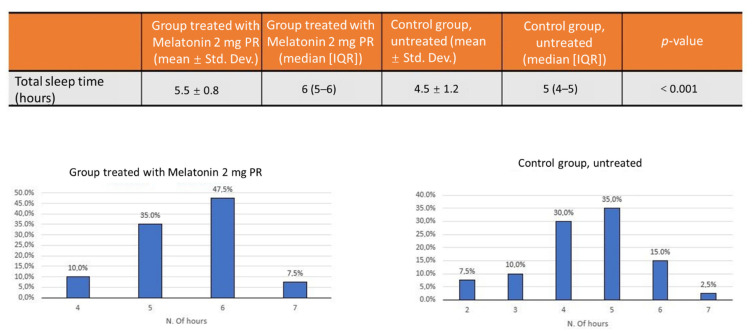
The group of patients treated with PRM 2 mg had a median of 6 h of sleep (47.5%), compared with a median of 5 h (35%) in the untreated group.

**Figure 3 jcm-10-05857-f003:**
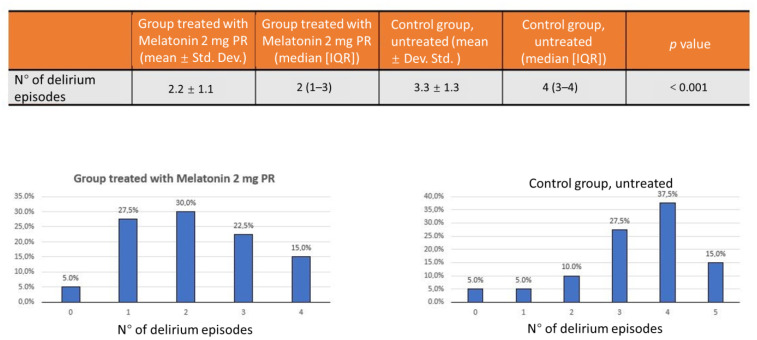
Significant reduction of episodes of delirium in the group treated with melatonin PRM 2 mg: 30% episodes in the median versus 37.5% episodes in the untreated group.

**Figure 4 jcm-10-05857-f004:**
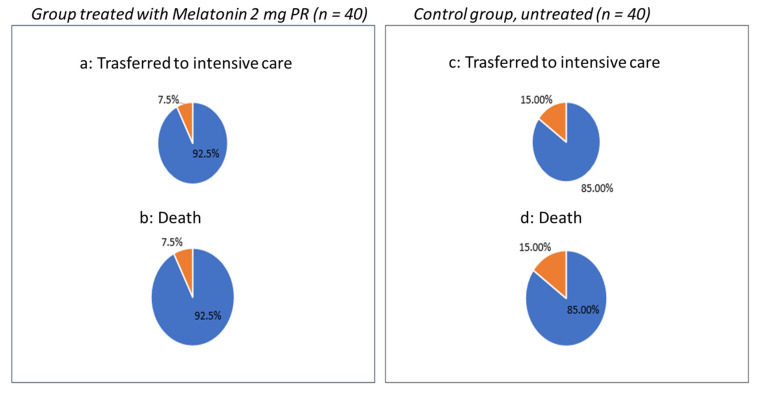
Number of patients transferred from sub-intensive to intensive care: in the group treated with PRM 2 mg, only 7.5% (**a**) versus 15% (**c**) of the un-treated group were transferred to intensive care due to clinical worsening, and 15% of the untreated group (**d**) died versus 7.5% of the treated group (**b**).

**Table 1 jcm-10-05857-t001:** Clinical and blood parameters of two groups.

Group Treated with Melatonin 2 mg PR (*n* = 40)			Control Group Untreated (*n* = 40)		
Clinical parameters	Mean	Std. Dev.	Clinical parameters	Mean	Std. Dev.
BMI	29.1	2.6	BMI	29.8	4.0
Creatinine (mg/dL)	1.4	0.4	Creatinine (mg/dL)	1.6	0.6
Azotemia (mg/dL)	64.5	22.9	Azotemia (mg/dL)	87.0	42.9
Hemoglobin (mg/dL)	11.0	1.2	Hemoglobin (mg/dL)	11.0	1.8
Glycemia (mg/dL)	112.6	30.5	Glycemia (mg/dL)	142.5	58.4
GOT (UI/L)	48.2	32.2	GOT (UI/L)	44.8	16.0
GPT (UI/L)	47.8	27.8	GPT (UI/L)	47.4	13.3
PCR (mg/dL)	19.5	15.6	PCR (mg/dL)	15.7	11.4

BMI: Body Mass Index; GOT: glutamate oxaloacetate transaminase; GPT: glutamate pyruvate transaminase; PCR: C Reactive Protein.

**Table 2 jcm-10-05857-t002:** Comparison between the two groups on the number of days of total Hospitalization, days of subintenive department, days spent in non invasive ventilation and days with high flow oxygen ventilation.

Days	Group Treated with Melatonin 2 mg PR (Mean ± Std. Dev.)	Group Treatd with Melatonin 2 mg PR (Median QR)	Control Group Untreated (Mean ± Dev. Std.)	Control Group Untreated (Median IQR)	*p*-Value
Hospitalization	31.3 ± 6.8		34.3 ± 6.9		0.03
In subintensive care	12.3 ± 3.2		20.1 ± 6.1		<0.001
With NIV Non Invasive Ventilation	5.2 ± 3.0	5 (3–7.3)	12.5 ± 4.2	11 (10–15.3)	<0.001
With High flow oxygen ventilation	7.1 ± 2.5	7 (5–8)	7.7± 3.2	7 (5–10)	0.436

NIV, non-invasive ventilation; IQR, interquartile range; PR, Prolonged-Release.

## Data Availability

All data, models, or codes generated or used during the study are available from the corresponding authors by request.

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
