# Peer review of "Efficacy of Prolonged-Release Melatonin 2 mg (PRM 2 mg) Prescribed for Insomnia in Hospitalized Patients for COVID-19: A Retrospective Observational Study"

_jcm, 2021, doi:10.3390/jcm10245857_

Round 1

Reviewer 1 Report

  1.  There are many English language corrections and style notes needed for this manuscript.  Normally, I would make suggestions and corrections myself but the amount of editing required is really extensive.  
  2. In your results section, the text includes many instances of numbers that do not have accompanying units. You do not indicate whether you are talking about hours or minutes for sleep, days or hours for ventilation or 
  3. Commas need to be replaced by decimal points for numbers.
  4. When comparing groups of people, you include some risk factors like BMI but other major risk factors such as diabetes and hypertension are not noted.
  5. You note that the patients enrolled had a greater rate of progression towards non-invasive ventilation but were controls sicker than the treatment group to begin with.  Were standardized parameters used to determine which patients stayed in the subacute unit or progressed to the intensive care unit?  Similarly, were there standardized parameter used for implementation of ventilatory support?
  6. How were delirium episodes defined?  I cannot see a standardized definition that is used in the study.

Reviewer 2 Report

Title:
Title of this manuscript should be identify as retrospective observational study
Introduction
1. The objective of this study is totally unclear (Line number: 51-54). 
2. Most recent in randomized controlled trials Mousavi et al. demonstrated that the oral melatonin tablets could substantially improve sleep quality in hospitalized COVID-19 patients (DOI: 10.1002/jmv.27312). Hence, significance of this work downgraded. 
3. Line number (43): "PRM 2 mg showed to mimic the physiological release of melatonin by releasing melatonin gradually and acting on melatonin receptors." relevant citation missing. 
4. Line number (45): "PRM 2 mg has been shown to be effective in improving sleep latency (SOL), Total Sleep 45 time (TST) without altering the physiological sleep structure." relevant citation missing. 

Materials and Methods
1. Where authors conducted and collected the data this retrospective observational study is unclear. 
2. Authors should be mentioned explicitly about which Institutional Review Board approved this study.
3. How insomnia was diagnosed is unclear.

Results:
1. Data representation too complicated. 
2. Authors did not carefully prepared the results. For example, Figure-4, "Transferred" spelling mistake.  

Conclusions:
1. Line number 171: "as well as a potential anti-inflammatory and antioxidant effect" this part should be deleted. 

Round 2

Reviewer 2 Report

The authors sincerely revised the manuscript as per reviewer suggestion and recommendation. Now, quality of this manuscript significantly improved. I endorse current form of the manuscript for publication. Thanks